# Prevalence of lumbar disc herniation and its associated factors: A cross-sectional study in Gansu

Zhiwei Chen[1,2], Jirong Zhao[1]*, Li Wang[1], Hong Shao[1], Liangjia Cao[1], Xueyun He[1], Zhenghan Yang[1], Junfei Ma[1], Qianwen Chen[1], Peng Jiang[1], Licun Zhang[1], Jihong Hu[1]*

**1** Public Health School, Gansu University of Chinese Medicine, Lanzhou, China, **2** Gansu Provicial Hospital of TCM, Lanzhou, China

☯ These authors contributed equally to this work.

\* 1325908320@qq.com (JZ); hujihonghappy@163.com (JH)

## Abstract

### Background and objective

The prevalence rate of LDH and its influencing factors in Gansu is unclear. This study aims to analyze the prevalence of LDH and influencing factors in Gansu.

### Methods

A stratified multi-stage random sampling method was used to obtain representative samples of residents more than 18 years old from <1500m, 1500-3500m, and >3500m altitude sites in Gansu, China, in June 2022 to August 2022. A unified questionnaire was used for the investigation.

### Results

The total number of people surveyed was 4545, with a prevalence rate of 22.77% for LDH. LDH prevalence differed by latitude, city, age, sex, nationality, education, marital status, income, drinking habits, residence duration, physical activity, exercise intensity, health status (including hypertension, diabetes, CHD, chronic infection, and tumors) (all $P$ <0.05). Logistic regression showed higher LDH risk at high (OR = 2.250) and middle latitudes (OR = 2.551), among males (OR = 0.808), ages 35-49 (OR = 1.530) and 50-64 (OR = 1.991), Tibetans (OR = 1.533) and Hui (OR = 0.557), alcohol consumers (OR = 0.696), those with moderate (OR = 0.742) and mild physical activity (OR = 0.840), cadres (OR = 0.46), workers (OR = 1.568), farmers/herdsmen (OR = 1.840), married individuals (OR = 2.239), residents for ≥36 months (OR = 0.618), annual income ≥50,000 yuan (OR = 1.246), central obesity (OR = 1.399), and those with tumors (OR = 3.953), hypertension (unknown, OR = 1.817), diabetes (yes, OR = 0.529, unknown, OR = 0.565), CHD (unknown, OR = 1.404), and osteoporosis (unknown, OR = 1.404).

**Data Availability Statement:** All relevant data are within the manuscript and its supporting information files.

**Funding:** Science and technology projects of Gansu, Grant Numbers. 21ZD4FA009.

**Competing interests:** All authors of this research paper have no conflicts of interest to declare.

## Conclusion

The prevalence of LDH was high, varying significantly with latitude, gender, and nationality, indicating potential lifestyle and demographic influences in Gansu.

## 1. Introduction

Lumbar disc herniation (LDH) is one of the common causes of chronic low back pain (LBP) and sciatica [1], the basis of unchanged spinal disorders, and the most common cause of disability in community [2]. According to the data of the World Health Organization (WHO), the main symptoms of LDH are low back pain and sciatica, which has a high prevalence and disability rates in both developing and developed countries [3], about 2/3 of adults have experienced low back pain [4], and about 85% of LDH patients have sciatica [5]. The global average prevalence and incidence of LDH are 14% to 20%, 2% to 4%, respectively, mostly in adults aged 30~50 years and occur at $L_{4\sim5}$ and $L_5 \sim S_1$, males more often in females [6].

At such a high percentage of LDH prevalence in the global population, LBP due to degenerative disc disease (e.g., LDH) has become the most common complaint in primary care and the second most common cause of chronic pain [7]. Due to the long duration of LDH, patients' suffering and severe paralysis, LDH has brought many adverse effects on patients' work and life [8], as well as a heavy burden on society [9]. Meanwhile, as LDH accounts for an increasing proportion of the disease burden and healthcare expenditures, we need to invest more in research and development to identify new and more effective intervention strategies [10].

The number of patients with lumbar spine disease in China is more than 200 million, of which about 15.2% are LDH patients [11]. Gansu is situated in Northwest China and is characterized by its complex geographic location and variable climate. Additionally, the region experiences unbalanced socioeconomic development. Therefore, it is crucial to study the risk factors for LDH prevalence in this region. Unfortunately, the current detailed epidemiologic information on LDH in Gansu Province is insufficient. There are three articles reported about the prevalence of LDH in Zhangye 16.28% [12], and Tianshui 12% [13], and Dingxi [14]. Above all studies only reported the prevalence of LDH in the local area of Gansu. At present, there are no studies on influencing factors of LDH in Gansu. In addition, Gansu is also an area inhabited by ethnic minorities, and this factor may also be a potential factor influencing the prevalence of LDH. Therefore, we hope that this study will effectively control the risk factors of LDH in Gansu, provide detailed data on the various factors affecting LDH, and provide effective support for the development of LDH prevention strategies to reduce the prevalence of LDH.

## 2. Materials and methods

### 2.1 Study population

A cross-sectional study on LDH was conducted among adults in Gansu from June 2022 to August 2022. The study was approved by ethics committee of the Gansu Provincial Hospital of Chinese Medicine (Lanzhou, China) (2021-119-01), and written informed consent was obtained from participants before the investigation. The sample size was calculated through an online-software, OpenEpi, with a 95% confidence interval (CI), anticipated prevalence rate of LDH (20%), and 3.0 design effects due to latitude levels. A stratified multi-stage random sampling method was used to select 5 medical institutions that had a good foundation for research

cooperation with us at high (n = 1), middle (n = 3), and low (n = 1) altitudes in Gansu Province, with a predicted sample of 1600, considering a 10% non-response rate, and 1800 local residents at different altitudes were selected as study subjects, totaling 5400 residents.

Sampling details were as follows: We conducted a stratified, multistage sampling in three randomly selected counties based on altitude: <1500m, 1500m-3500m, and > = 3500m. Native residents aged ≥18 years who had lived locally for at least 6 months were included. Initially, we selected Zhangye City (<1500m), Linxia City, Tongwei County, Zhuoni County (1500m-3500m), and Maqu County (> = 3500m). In the second stage, ten villages from each of these five counties were randomly selected.

Eligible participants for this survey are those without mental illness. Exclusion criteria include: (1) low back and leg pain from other causes (e.g., trauma, renal or ureteral disease, chronic pelvic inflammatory disease); (2) recent hospital admissions for acute illnesses (e.g., ureteral stones [15], lumbar sprains [16]). LDH patients were identified through medical history, symptoms, and CT/MRI results [17]. A total of 4545 randomly sampled residents participated in this survey, including 1035 LDH patients.

## 2.2 Data collection and definition

We use a unified, standardized electronic questionnaire (data collection management system for LDH) to collect some information by face-to-face interviews. Collected information includes demographic information, past medical history, and lifestyle risk factors. The demographic information includes gender (male or female), age (<35 years, 35-49 years, 50-64 years, ≥ 65 years), nationality (Han, Tibetan, Hui, other), spouse status, education level (primary and below, middle school, high school and above), occupation (cadre, worker, unemployed, farmer and herdsman), residence time (6 months, 24 months, ≥36 months), and annual incoming (<3000 yuan/year, 3000–yuan/year, ≥5000 yuan/year). The past medical history includes hypertension, diabetes, CHD, stroke, central obesity, obesity, chronic infectious diseases, and tumors. The lifestyle risk factors include physical activity (severe, moderate, mild), exercise intensity (severe, moderate, mild), current smoking ("yes" is defined by smoked ≥100 cigarettes in their lifetime and smoked in the last 28 days), and current drinking (answers range from least 300g of beer or 40g of liquor or 125g of fruit wine consumed in the past 30d), state of health, which was judged by medical history including hypertension, diabetes, dyslipidemia, CHD, osteoporosis, cerebral apoplexy, chronic infectious diseases, tumors, and so on. All investigators were trained before the survey. Central obesity, as assessed by waist circumference, was defined at waist circumference > = 90 cm in men or > = 80 cm in women. Body mass index (BMI) was categorized as low weight (BMI<18.5), normal weight (BMI 18.5-23.99), overweight (BMI> = 24), and obesity (BMI> = 28) [18].

## 2.3 Statistical methods

Normally distributed continuous were described as averages (standard range), and non-normally distributed continuous were described as medians (interquartile range, IQR). Categorical variables were described as absolute numbers and frequencies. T test or Mann–Whitney U-tests were used for comparisons among means. A Chi-square test was used for comparisons among categorical variables. Logistic regression models were used to examine the associations between LDH and the risk of LDH, which were estimated by OR with 95% CI and presented with Forest Figure. All analyses were conducted using SPSS version 25.0 (Chicago, IL) with the significance level set at a two-tailed $P<0.05$.

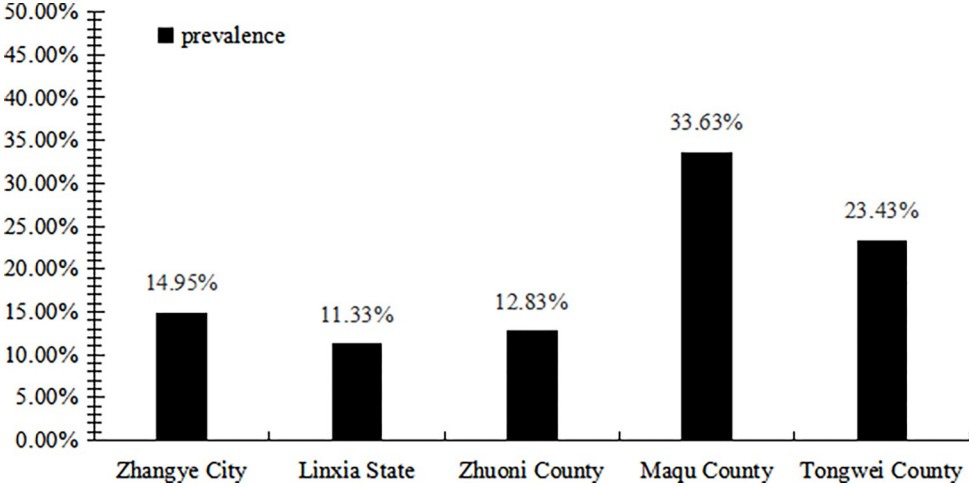

**Fig 1. Prevalence of LDH in each city.** Maqu county is located in high-altitude areas (> = 3500 m); Linxia city, and Tongwei counties and Zhuoni counties are located in middle altitude (1500 m-3500 m); Zhangye City is located in low altitude (< = 1500m). The prevalence of LDH significantly increased with altitudes (*P<0.05*), and significantly different among 5 cities or counties (*P<0.05*).

## 3. Results

### 3.1 The prevalence of LDH was different latitudes and cities

The prevalence of LDH in Gansu was 22.77%. The prevalence of LDH at different altitudes (respectively, low 14.95%, middle 23.10%, high 28.28%) was statistically significant ($\chi^2$ =101.96, *P<0.05*) and increased with altitudes ($\chi_{trend}^2$ = 91.96, *P<0.05*). The prevalence of LDH in 5 cities was significantly different (all *P* <0.05), furthermore, in Maqu was highest (33.63%), and secondly, in Tongwei (23.43%) (Fig 1).

### 3.2 Comparative analysis of prevalence of LDH in different groups

The prevalence of LDH in different groups was statistically significant (all *P* <0.05), including age, sex, annual incomes, nationality, education level, occupation, length of residence, and marital status groups (Table 1); and also significantly different in different lifestyle groups (all *P* <0.05), including physical activity, exercise, and drinking (Table 2); and discrepancies were significant in different medical history groups (all *P* <0.05), including hypertension, diabetes, CHD, chronic infectious diseases, and tumors (Table 3).

### 3.3 Logistic regression analysis of factors influencing LDH

The results of logistic regression analysis showed that, compared with low latitude, female, age <35 yeas, other nationality, without spouse, primary school or lower, unemployment, residence period 6-12 months, annual income < 10,000 yuan, mild physical activity, no drinking, no central obesity, no obesity and no related medical history, latitude (high latitude, OR = 2.250; middle latitude, OR = 2.551), age group (35-49 years, OR = 1.530; 50-64 years, OR = 1.991), Tibetan (OR = 1.533), occupation (worker, OR = 1.568; farmer&herdsman, OR = 1.840), with spouse (OR = 2.239), annual income >=50,000 yuan (OR = 1.246), central obesity (OR = 1.399), hypertension (unknown, OR = 1.817), CHD (unknown, OR = 1.404), tumor (unknown, OR = 3.953), and Osteoporosis (unknown, OR = 1.404) were associated with the increased risk of LDH. However, the male (OR = 0.808), Hui ethnic (OR = 0.557), physical activity (moderate, OR = 0.742; mild, OR = 0.840), cadre (OR = 0.46), high school or

**Table 1. Prevalence of LDH in different demographic groups.**

| | LDH | | | |
|---|---|---|---|---|
| | Number of LDH (n = 1035) | Prevalence (%) | χ2 | P |
| **Age group** | | | **31.78** | **<0.001** |
| <35(n = 1514) | 283 | 18.69 | | |
| 35-49(n = 1179) | 306 | 25.95 | | |
| 50-64(n = 1408) | 362 | 25.71 | | |
| ≥65(n = 444) | 84 | 18.92 | | |
| Gender | | | 16.09 | <0.001 |
| Male(n = 2026) | 405 | 19.99 | | |
| Female(n = 2519) | 630 | 25.00 | | |
| Nationality | | | 75.01 | <0.001 |
| Han(n = 2598) | 517 | 19.90 | | |
| Tibetan(n = 1404) | 430 | 30.63 | | |
| Hui(n = 419) | 70 | 16.71 | | |
| Others(n = 124) | 18 | 14.52 | | |
| marital status | | | 47.69 | <0.001 |
| Yes(n = 3915) | 959 | 24.50 | | |
| No(n = 630) | 76 | 12.06 | | |
| Educational level | | | 32.81 | <0.001 |
| Primary school or lower(n = 1697) | 459 | 27.05 | | |
| Middle school(n = 1071) | 240 | 22.41 | | |
| High school or higher(n = 1777) | 336 | 18.91 | | |
| Occupation | | | 19.17 | <0.001 |
| Cadre(n = 92) | 11 | 11.96 | | |
| Worker(n = 3887) | 913 | 23.49 | | |
| Unemployed(n = 329) | 50 | 15.20 | | |
| Farmer and herdsman (n = 237) | 61 | 25.74 | | |
| length of residence | | | 43.20 | <0.001 |
| 6- months(n = 238) | 76 | 31.93 | | |
| 24- months(n = 343) | 118 | 34.40 | | |
| ≥36 months(n = 3964) | 841 | 21.22 | | |
| annual income | | | 9.92 | 0.007 |
| ≤30,000 RMB(n = 1472) | 316 | 21.47 | | |
| 30,000- RMB(n = 1626) | 348 | 21.40 | | |
| ≥50,000 RMB(n = 1447) | 371 | 25.64 | | |

higher (OR = 0.629) residence time >36 months (OR = 0.574), alcohol consumption (OR = 0.696), and diabetes (yes, OR = 0.529; unknown, OR = 0.565) were related with the reduced risk of LDH (Fig 2).

## 4. Discussion

LDH is one of the common causes of chronic LBP, which seriously reduces the quality of life of patients as a chronic painful disease with a long recovery time and poor curative effect. Lumbar spine diseases are as high as the fifth in the medical costs of various diseases in hospitals, and the costs incurred by absenteeism and disability caused by lumbar spine diseases are higher than those of other diseases, causing heavy mental pressure and economic burden to patients [19]. Gansu has unique geographic characteristics and dietary habits, and the

**Table 2. Prevalence of LDH in different lifestyle.**

| | LDH | | | |
|---|---|---|---|---|
| | Number of LDH patients(n = 1035) | Prevalence rate(%) | χ2 | P |
| **physical activity** | | | **18.24** | **<0.001** |
| severe(n = 755) | 208 | 27.55 | | |
| moderate(n = 1504) | 296 | 19.68 | | |
| mild(n = 2286) | 531 | 23.23 | | |
| exercise intensity | | | 17.56 | <0.001 |
| severe(n = 413) | 92 | 22.28 | | |
| moderate(n = 574) | 92 | 16.03 | | |
| mild(n = 3558) | 851 | 23.92 | | |
| current smoking | | | 0.27 | 0.607 |
| Yes(n = 579) | 127 | 21.93 | | |
| No(n = 3966) | 908 | 22.89 | | |
| current drinking | | | 10.01 | 0.002 |
| Yes(n = 574) | 101 | 17.60 | | |
| No(n = 3971) | 934 | 23.52 | | |

prevalence of LDH and its influencing factors vary in different regions and ethnic groups, resulting in a lack of large-scale epidemiologic studies on LDH prevalence in Gansu. Therefore, analyzing and exploring the prevalence of LDH in Gansu is of great significance for the control and prevention of LDH.

The results of the study showed that the prevalence of LDH in the resident population of Gansu Province was 22.77%, which exceeded the nation average [10], and there were large differences among the findings of LDH prevalence in five different areas. It is certain that the prevalence of LDH may be related to the altitude at which the residents live, in addition to the different ethnic groups, regions, ages, and occupations of the respondents [20].

Furthermore, the results of multifactorial logistic regression analysis in Fig 2 showed that age, gender, ethnicity, spouse, education, occupation, residential altitude, duration of residence, and annual household income were associated with risk of LDH in the residents of Gansu Province, which was inconsistent with the results of previous studies. According to the literature, LDH is twice as prevalent in men as in women and is most common between the ages of 30-59 years old [21,22]. Table 1 shows in detail the results of the influence of the basic information of the surveyed population on the prevalence of LDH. For age, we considered <30, 30-59, >=60 age groups with the prevalence of 17.51%, 25.28%, and 20.39%. The prevalence of LDH was higher in the female and age group of 35-64 years in Gansu at different ages, which may be related to the high work intensity, economic, and life stress of this age group in the region. It is interesting to pay attention to the fact that the prevalence of LDH was found to be 42.5% in women in the age group of > = 50 years in this study, which is also consistent with the high prevalence of LDH in middle-aged patients reported in the literature [23]. Women in this age group are in menopause; estrogen and luteinizing hormone tend to fluctuate in the female body, and the level of sex hormone production is significantly reduced compared to the previous period, and the effect of hormone levels is a risk factor for osteoporosis [24]. It was found that the prevalence of osteoporosis in men and women aged 50 years and above was 6.46% and 29.13% [25], respectively, and osteoporosis is a risk factor for LDH, which contributes to the higher prevalence of LDH in women than in men in this study. Our findings suggest an increased prevalence of LDH in patients who are unaware of their osteoporosis symptoms, which may be due to the fact that patients do not intervene in the treatment of

**Table 3. Comparison of prevalence of LDH in medical status.**

| | LDH | | χ2 | P |
|---|---|---|---|---|
| | Number of LDH (n = 1035) | Prevalence (%) | | |
| hypertension | | | 7.75 | 0.021 |
| Yes(n = 369) | 79 | 21.41 | | |
| No(n = 3450) | 762 | 22.09 | | |
| Unknown(n = 726) | 194 | 26.72 | | |
| diabetes | | | 12.87 | 0.002 |
| Yes(n = 181) | 24 | 13.26 | | |
| No(n = 3453) | 820 | 23.75 | | |
| Unknown(n = 911) | 191 | 20.97 | | |
| hyperlipidemia | | | 0.39 | 0.824 |
| Yes(n = 127) | 27 | 21.26 | | |
| No(n = 3430) | 788 | 22.97 | | |
| Unknown(n = 988) | 220 | 22.27 | | |
| osteoporosis | | | 0.12 | 0.940 |
| Yes(n = 84) | 20 | 23.81 | | |
| No(n = 3455) | 783 | 22.66 | | |
| Unknown(n = 1006) | 232 | 23.06 | | |
| coronary heart disease | | | 6.87 | 0.032 |
| Yes(n = 43) | 14 | 32.56 | | |
| No(n = 3557) | 831 | 23.36 | | |
| Unknown(n = 945) | 190 | 20.11 | | |
| cerebral apoplexy | | | 1.29 | 0.525 |
| Yes(n = 32) | 9 | 28.13 | | |
| No(n = 3603) | 829 | 23.01 | | |
| Unknown(n = 910) | 197 | 21.65 | | |
| central obesity | | | 0.83 | 0.362 |
| Yes(n = 1167) | 277 | 23.74 | | |
| No(n = 3378) | 758 | 22.44 | | |
| obesity | | | 0.177 | 0.674 |
| Yes(n = 249) | 54 | 21.69 | | |
| No(n = 4296) | 981 | 22.84 | | |
| chronic infectious diseases | | | 6.21 | 0.013 |
| Yes(n = 61) | 22 | 36.07 | | |
| No(n = 4484) | 1013 | 22.59 | | |
| tumor | | | 46.40 | <0.001 |
| Yes(n = 76) | 42 | 55.26 | | |
| No(n = 4469) | 993 | 22.22 | | |

LDH when they are unaware of their condition. In this study, we also found that the prevalence of LDH was related to yearly incoming, which reflected economic conditions, and the willingness of the population to participate in routine physical examination is stronger, thus diagnosing more LDH patients.

In addition, spousal status also increased the risk of LDH, which may be related to the fact that married people tend to take on more domestic work and childbearing in family life [26]. Heavy laborers [27], such as workers or farmers and herdsman, have a higher incidence of LDH because this group of people often need to bend over repeatedly and their lower back muscles are more susceptible to fatigue and injury [28]. We also found that moderate and severe physical

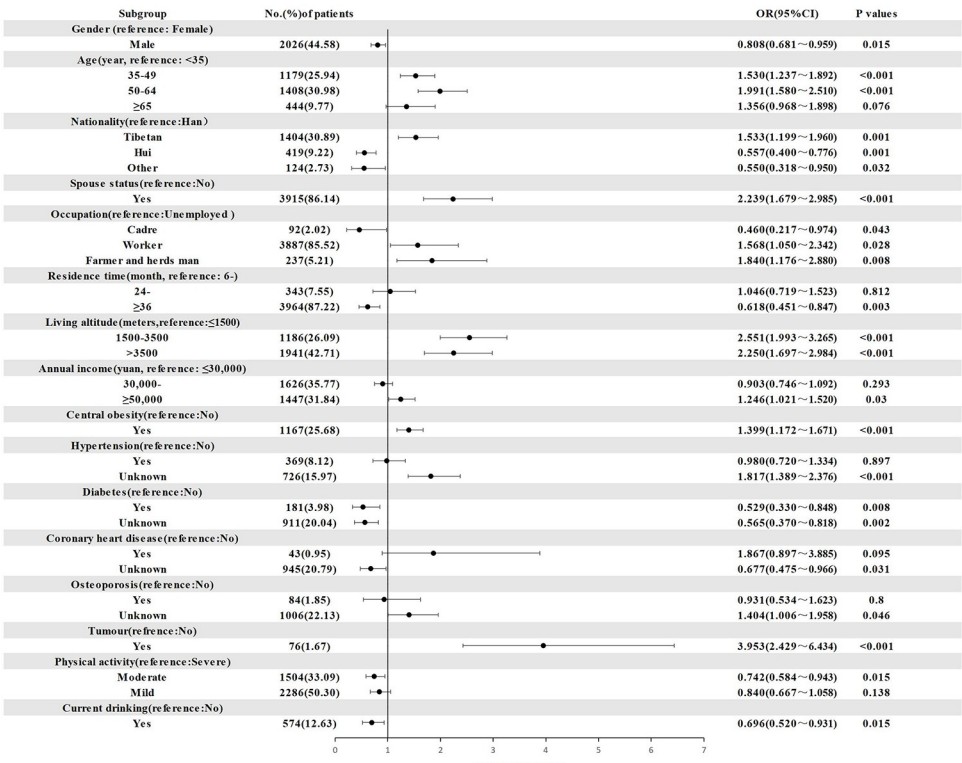

Variables including gender, age (≤35 years, 35-49 years, 50-64 years, ≥65 years), nationality (Han, Tibetan, Hui, Others), spouse status (yes or no) educational levels (primary school or lower, middle school, high school or higher), occupation (cadre, worker, unemployed, others), residence time(6-months,24-months, ≥ 36months),living altitude(≤ 1500meters,1500-3500 meters,>3500meters), annual incoming (<3000 yuan/yr, 3000-yuan/yr, or≥5000 yuan/yr), and past medical history (including hypertension, diabetes, hyperlipidemia, osteoporosis, coronary heart disease, cerebral apoplexy, centra obesity, obesity, chronic infectious diseases and tumor), lifestyle risk factors [physical activity (severe, moderate, mild), exercise intensity (severe, moderate mild), current smoking ("yes" is defined by smoked ≥100 cigarettes in their lifetime and smoked in the last 28 days), and current drinking (answers range from least 300g of beer or 40g of liquor or 125g of fruit wine consumed in the past 30d).

**Fig 2. Logistic regression analysis of the risk factors associated with LDH.**

activity was a risk factor for LDH compared to heavy physical activity, whereas light physical activity would not contribute to the development of LDH. The plateau environment to which the residents of Gansu belong is also an important factor influencing the prevalence of LDH in the region compared with the traditional factors affecting LDH in the past [12,13]. Residents at high altitudes have long winters, short summers, and relatively cold weather compared to those living at lower altitudes, so they tend to be sedentary indoors and lack outdoor exercise [29]. Tibetans are related with the elevated risk of LDH, which may be related to the special nomadic lifestyle and geographical environment of ethnic minorities environment, backward economic and cultural level, high intensity of physical labor, and other factors [30].

Although LDH is never caused by one factor alone, Huang et al. [31] found that smoking promotes LDH, and Andersen et al. [32] found that smoking was associated with a higher risk of reoperation due to LDH. We found that smoking was not a risk factor for LDH, which may be related to the survey sample size. Central obesity and BMI both are the risk factors for LDH [33]. Bostman OM [34] points to an elevated BMI as an important factor contributing to the significant increase in the incidence of LDH, with central obesity patients experiencing an increased burden on the muscles of the low back, which in turn increases intervertebral disc pressures and ultimately leads to the development of LDH. The relationship of central obesity and LDH was founded in this study too, after adjusted for central obesity and BMI in multivariable Logistic regression. It suggested that central obesity played a more important role on LDH than BMI. Studies have found that drinking alcohol can increase the risk of developing

LDH, and thus current drinking is also a risk factor for LDH [35]. However, we found that current alcohol consumption is related to the reduced risk for developing LHD.

In addition, li et al. [36] found that some chronic diseases, such as diabetes mellitus (DM) and hypertension, are also related to the occurrence of LDH. However, in this study, it was found that diabetes mellitus decreased the risk of LDH. Additionally, the effect of tumor disease on LDH was also investigated in this study. Relevant literature reports no association between tumor and LDH disease, but tumors compress the intervertebral disc nerves, leading to an increased incidence of LDH [37]. Our study found tumors to be an important risk factor for LDH. Generally, it should be noted that patients with chronic diseases such as osteoporosis, hypertension, DM, and CHD were not sure whether they suffered from chronic diseases due to the early stage of the patients, who do not have obvious symptoms. Until identified, the patients will be medicated for the intervention. The prevalence of LDH is significantly higher in patients who do not know whether they have hypertension or osteoporosis in this study.

All in all, the prevalence of LDH is high, and the rate was different in different latitudes, genders, and nationalities, which probably led to the discrepancies of LDH on lifestyles, demographic characteristics groups, and disease among Gansu residents. In addition to uncontrollable factors such as age, nationality, and gender, it is necessary to avoid moderate to heavy physical activities to reduce the occurrence of LDH. At the same time, health education and consciousness of the local population should be strengthened to be able to do regular medical checkups to prevent chronic diseases such as osteoporosis, tumor, hypertension, diabetes, CHD, etc., and reduce the risk of LDH. The main strength is that this is the first epidemiological study based on populations at different altitudes in Gansu, focusing on exploring the influencing factors of LDH. In addition, we developed strict inclusion criteria to select subjects, standardized data collection methods and well-designed statistical analysis methods, which reduced detection bias and made the study results more convincing. Despite some positive findings in this study, several limitations that could not be avoided. Firstly, despite our rigorous training of investigators prior to investigation, there can be information biases at the time of investigation, which can affect our results. Secondly, a larger sample survey should be needed, which may be useful to find the positive effects of smoking and BMI on LDH.

## 5. Conclusions

The study finds that Gansu has a 22.77% prevalence of LDH, higher than the national average, with a notable rise at higher altitudes. Prevalence varies by demographics, lifestyle, and chronic conditions. Increased risk factors include high altitude, middle and older age, Tibetan ethnicity, specific occupations (workers, farmers/herders), marital status, higher income, central obesity, and medical histories of hypertension and tumors. Conversely, being male, of Hui ethnicity, moderately active, well-educated, a long-term resident, an alcohol consumer, and diabetic is associated with a lower risk. The study emphasizes the need to reduce moderate to heavy physical activity and highlights the importance of enhanced health education and regular medical check-ups to prevent chronic diseases related to LDH.

## Supporting information

**S1 Data.**
(XLSX)

## Author Contributions

**Conceptualization:** Jirong Zhao, Jihong Hu.

**Data curation:** Li Wang, Xueyun He, Zhenghan Yang, Peng Jiang.

**Formal analysis:** Liangjia Cao, Zhenghan Yang, Peng Jiang.

**Methodology:** Zhiwei Chen, Li Wang, Hong Shao, Junfei Ma, Licun Zhang.

**Visualization:** Li Wang, Qianwen Chen.

**Writing – original draft:** Jihong Hu.

**Writing – review & editing:** Jirong Zhao, Li Wang, Jihong Hu.

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
