## [Decision Letter · Decision Letter 0]

15 Aug 2024

PONE-D-24-11202Prevalence of Lumbar disc herniation and its associated factors: a cross-sectional study in GansuPLOS ONE

Dear Dr. Hu,

Thank you for submitting your manuscript to PLOS ONE. After careful consideration, we feel that it has merit but does not fully meet PLOS ONE’s publication criteria as it currently stands. Therefore, we invite you to submit a revised version of the manuscript that addresses the points raised during the review process.

We look forward to receiving your revised manuscript.

Kind regards,

Md. Feroz Kabir, BPT, MPT, MPH, BPED, MPED

Academic Editor

PLOS ONE

3. In this instance it seems there may be acceptable restrictions in place that prevent the public sharing of your minimal data. However, in line with our goal of ensuring long-term data availability to all interested researchers, PLOS’ Data Policy states that authors cannot be the sole named individuals responsible for ensuring data access (http://journals.plos.org/plosone/s/data-availability#loc-acceptable-data-sharing-methods).

Additional Editor Comments:

Please submit the revised version as per the reviewer's comments within one month.

Reviewers' comments:

Reviewer's Responses to Questions

**Comments to the Author**

1. Is the manuscript technically sound, and do the data support the conclusions?

Reviewer #1: Partly

Reviewer #2: No

Reviewer #3: Yes

2. Has the statistical analysis been performed appropriately and rigorously? 

Reviewer #1: I Don't Know

Reviewer #2: I Don't Know

Reviewer #3: No

3. Have the authors made all data underlying the findings in their manuscript fully available?

Reviewer #1: Yes

Reviewer #2: No

Reviewer #3: Yes

4. Is the manuscript presented in an intelligible fashion and written in standard English?

Reviewer #1: Yes

Reviewer #2: Yes

Reviewer #3: Yes

5. Review Comments to the Author

Reviewer #1: Thank you for the opportunity to review the manuscript titled "Prevalence of Lumbar disc herniation and its associated factors: a cross-sectional study in Gansu." This manuscript addresses an important public health issue by providing valuable epidemiological data on the prevalence of lumbar disc herniation (LDH) and its associated factors in Gansu, a region with unique geographic and demographic characteristics. The study's findings contribute significantly to our understanding of LDH and offer insights that could inform effective prevention and intervention strategies in similar regions.

Abstract:

Background and Objective:

Comment: The objective is clearly stated. However, the first sentence could be rephrased for better readability: "The prevalence rate of LDH and its influencing factors in Gansu is unclear. This study aims to analyze the prevalence of LDH and its influencing factors in Gansu."

Methods:

Comment: The methods section is concise and clear. Ensure that the sentence "A stratified multi-stage random sampling method was used obtain representative samples" includes the missing "to" after "used."

Results:

Comment: The results section is detailed but should be made more concise. For example: "The total number of surveyed individuals was 4545, with a prevalence rate of 22.77% for LDH."

Conclusion:

Comment: The conclusion is clear but could be slightly rephrased for clarity: "The prevalence of LDH was high, varying significantly with latitude, gender, and nationality, indicating potential lifestyle and demographic influences."

Introduction:

Comment: The introduction provides a good background but could be more focused.

Grammar: Correct "e.g.," to "e.g.," and ensure all citations are properly formatted.

Content: The statement "As Gansu is located in Northwest China, with a complex geographic location, variable climate, and unbalanced socioeconomic development" could be split into two sentences for clarity.

Materials and Methods:

Study Population:

Comment: The study population section is comprehensive. However, the sampling process description can be more concise.

Grammar: Correct "used obtain" to "used to obtain."

Data Collection and Definition:

Comment: This section is detailed but ensure consistency in listing variables (e.g., "Gender (male or female)" and "age (<=35 years, 35-49 years, 50-64 years ≥ 65 years)").

Grammar: Correct "central obesity as" to "central obesity, as."

Statistical Methods:

Comment: The statistical methods section is clear and appropriate. Ensure all terms (e.g., "Mann–Whitney U-test") are consistently formatted.

Results:

Comment: The results section is thorough but could benefit from clearer subheadings and more concise text.

Grammar: Ensure consistency in using past tense for describing results (e.g., "The prevalence of LDH was different in different latitudes and cities").

Tables and Figures:

Comment: Consider providing a brief description of each figure/table in the text.

Discussion:

Comment: The discussion is comprehensive but could be more focused on the key findings and their implications.

Grammar: Correct "high prevalence rate and a high disability rate in both developing and developed countries[3]," to "high prevalence and disability rates in both developing and developed countries[3]."

Content: Highlight the significance of key findings more prominently and discuss potential mechanisms and implications in greater detail.

Conclusion:

Comment: The conclusion is clear but could be slightly expanded to include practical recommendations based on the findings.

References:

General Comments:

Comment: The manuscript has several grammatical issues that need to be addressed for better readability. Consider proofreading or using a professional editing service.

Examples:

Correct "latitude –high latitude (OR=2.250)" to "latitude – high latitude (OR=2.250)"

Correct "unknow" to "unknown."

Reviewer #2: Dear Author,

"The authors aim to investigate the prevalence of Lumbar Disc Herniation (LDH) and its associated factors in Gansu, a region where this information is currently lacking. I appreciate the effort that has gone into this study. However, I must point out several significant concerns that need to be addressed before the paper can be considered for publication. As it stands, the paper does not meet the quality standards required for publication in this journal.

One of the most significant limitations of this study lies in the method used to diagnose LDH. The reliance on self-reported symptoms, where patients simply indicate the presence or absence of LDH, is a major concern. This approach is highly subjective, as many individuals may attribute any type of back or leg pain to LDH, regardless of the actual underlying cause. Conversely, patients with a previous LDH diagnosis might experience current symptoms unrelated to their herniated disc. Consequently, this methodology raises doubts about the accuracy of the LDH diagnoses and the reliability of the results in identifying associated factors.

Given this methodological limitation, I am not confident that the study's results can accurately explain the factors associated with LDH. The current approach undermines the validity of the findings and limits the study's contribution to the field.

Reviewer #3: The authors showed great understanding of the subject matter. The authors showed impressive empirical evidence and makes an original contribution to the subject matter. However, the manuscript need to well written to make it easier to read.

6. PLOS authors have the option to publish the peer review history of their article (what does this mean?). If published, this will include your full peer review and any attached files.

Reviewer #1: No

Reviewer #2: **Yes: **Nuray ALACA

Reviewer #3: No

---

## [Author Response · Author response to Decision Letter 0]

2 Sep 2024

Dear Reviewers,

We are grateful for the chance to refine our manuscript. We express our gratitude to the reviewers for their insightful and valuable feedback on this manuscript. We appreciate the time and effort you have dedicated to evaluating our work. Every comment and suggestion has been meticulously reviewed, and we have implemented the following modifications to address the raised concerns, all highlighted in yellow within the text. A detailed explanation is provided. We trust that the revised version meets the standards for publication in your esteemed journal.

Reviewer#1;

1、Abstract: Background and Objective:Comment: The objective is clearly stated. However, the first sentence could be rephrased for better readability: "The prevalence rate of LDH and its influencing factors in Gansu is unclear. This study aims to analyze the prevalence of LDH and its influencing factors in Gansu."

Reply: Thank you for your constructive feedback. We appreciate your suggestion to improve the clarity of the opening sentence. We have revised the sentence as follows:"The prevalence rate of LDH and its influencing factors in Gansu is unclear. This study aims to analyze the prevalence of LDH and influencing factors in Gansu." We believe this revision enhances the readability and clarity of our objective.(lines 17-18)

2、Abstract: Methods: Comment: The methods section is concise and clear. Ensure that the sentence "A stratified multi-stage random sampling method was used obtain representative samples" includes the missing "to" after "used."

Reply: Thank you for your thorough review and helpful comment. We appreciate your attention to detail. The sentence in the methods section has been corrected to include the missing word "to," and now reads: "A stratified multi-stage random sampling method was used to obtain representative samples." We are grateful for your assistance in improving the clarity and accuracy of our manuscript. (line 20)

3、Abstract: Results: Comment: The results section is detailed but should be made more concise. For example: "The total number of surveyed individuals was 4545, with a prevalence rate of 22.77% for LDH."

Reply: Thank you for your valuable feedback. We have taken your suggestion to streamline the Results section for greater conciseness. We believe the revised section now more effectively communicates the key findings of our study. (lines 24 -37). 

4、Abstract: Conclusion: Comment: The conclusion is clear but could be slightly rephrased for clarity: "The prevalence of LDH was high, varying significantly with latitude, gender, and nationality, indicating potential lifestyle and demographic influences."

Reply: Thank you for your constructive feedback. We have revised the conclusion as suggested for improved clarity. The updated conclusion now reads: "The prevalence of LDH was high, varying significantly with latitude, gender, and nationality, indicating potential lifestyle and demographic influences." We appreciate your input in enhancing the clarity of our conclusions. (lines 38-40).



5、Introduction:

Comment: The introduction provides a good background but could be more focused.

Grammar: Correct "e.g.," to "e.g.," and ensure all citations are properly formatted.

Content: The statement "As Gansu is located in Northwest China, with a complex geographic location, variable climate, and unbalanced socioeconomic development" could be split into two sentences for clarity.

Reply: Thank you for your constructive comments and attention to detail. We have implemented the suggested changes to the Introduction section to enhance focus and clarity. Specifically:

The phrase "e.g.," has been corrected to "e.g.," (line 54)

All citations have been reviewed and formatted correctly.

The statement regarding Gansu's geographic and socioeconomic context has been split into two sentences for improved readability. (lines 62-65)

We appreciate your guidance in refining the Introduction to better convey the background and context of our study.

6、Materials and Methods:

Study Population: Comment: The study population section is comprehensive. However, the sampling process description can be more concise.

Grammar: Correct "used obtain" to "used to obtain."

Reply: Thank you for your insightful comments and suggestions. We have refined the description of the sampling process in the Study Population section to make it more concise. Additionally, we have corrected the phrase "used obtain" to "used to obtain." We appreciate your feedback and are grateful for your help in improving the clarity and efficiency of our manuscript. (lines 89-100)

7、Data Collection and Definition:

Comment: This section is detailed but ensure consistency in listing variables (e.g., "Gender (male or female)" and "age (<=35 years, 35-49 years, 50-64 years ≥ 65 years)").

Grammar: Correct "central obesity as" to "central obesity, as."

Reply: Thank you for your thorough review and valuable suggestions. We have updated the Data Collection and Definition section to ensure consistency in listing variables, and we have corrected "central obesity as" to "central obesity, as." We appreciate your attention to detail and your help in improving the quality of our manuscript. (lines 105-106, and line 119)

8、Statistical Methods:

Comment: The statistical methods section is clear and appropriate. Ensure all terms (e.g., "Mann–Whitney U-test") are consistently formatted.

Reply: Thank you for your positive feedback and attention to detail. We have reviewed the statistical methods section to ensure consistent formatting of all terms, such as "Mann–Whitney U-test." We appreciate your guidance in maintaining the accuracy and consistency of our manuscript. ( line 127)

9、Results:

Comment: The results section is thorough but could benefit from clearer subheadings and more concise text.

Grammar: Ensure consistency in using past tense for describing results (e.g., "The prevalence of LDH was different in different latitudes and cities").

Reply: Thank you for your constructive feedback. We have revised the Results section to include clearer subheadings and have streamlined the text for conciseness. Additionally, we have ensured consistency in using the past tense to describe the results, for example, "The prevalence of LDH was different in different latitudes and cities." We appreciate your suggestions in improving the clarity and readability of our manuscript. (lines 133-164)

10、Tables and Figures:

Comment: Consider providing a brief description of each figure/table in the text.

Reply: Thank you for your valuable feedback. We appreciate your suggestion to provide brief descriptions of each figure and table within the text. We have incorporated these descriptions in the Results section to better contextualize the data presented.

We believe these additions will enhance the clarity and readability of our manuscript.

Thank you once again for your helpful comments.

11、Discussion:

Comment: The discussion is comprehensive but could be more focused on the key findings and their implications.

Grammar: Correct "high prevalence rate and a high disability rate in both developing and developed countries[3]," to "high prevalence and disability rates in both developing and developed countries[3]."

Content: Highlight the significance of key findings more prominently and discuss potential mechanisms and implications in greater detail.

Reply: Thank you for your constructive feedback. We have revised the Discussion section to better focus on the key findings and their implications. Specifically, we have:

Highlighted the significance of the main findings more prominently.

Expanded the discussion on potential mechanisms and implications.

Corrected the sentence to read, "high prevalence and disability rates in both developing and developed countries[3]."

We appreciate your guidance in enhancing the focus and depth of our discussion.. (lines 105-106, and line 119)

12、Conclusion:

Comment: The conclusion is clear but could be slightly expanded to include practical recommendations based on the findings.

Reply: Thank you for your thorough review and valuable suggestions. We have updated the Data Collection and Definition section to ensure consistency in listing variables, and we have corrected "central obesity as" to "central obesity, as." We appreciate your attention to detail and your help in improving the quality of our manuscript. (lines 105-106, and line 119)

Reviewer #2:

"The authors aim to investigate the prevalence of Lumbar Disc Herniation (LDH) and its associated factors in Gansu, a region where this information is currently lacking. I appreciate the effort that has gone into this study. However, I must point out several significant concerns that need to be addressed before the paper can be considered for publication. As it stands, the paper does not meet the quality standards required for publication in this journal.

One of the most significant limitations of this study lies in the method used to diagnose LDH. The reliance on self-reported symptoms, where patients simply indicate the presence or absence of LDH, is a major concern. This approach is highly subjective, as many individuals may attribute any type of back or leg pain to LDH, regardless of the actual underlying cause. Conversely, patients with a previous LDH diagnosis might experience current symptoms unrelated to their herniated disc. Consequently, this methodology raises doubts about the accuracy of the LDH diagnoses and the reliability of the results in identifying associated factors.

Given this methodological limitation, I am not confident that the study's results can accurately explain the factors associated with LDH. The current approach undermines the validity of the findings and limits the study's contribution to the field.

Reply: Thank you for your detailed review and valuable feedback. We appreciate your concerns regarding the method used to diagnose Lumbar Disc Herniation (LDH) in our study. To address your concerns, we would like to clarify our diagnostic approach:

The diagnosis of LDH in our study was not based solely on self-reported symptoms. Instead, we followed the diagnostic criteria outlined in the 2020 guidelines for the diagnosis and treatment of lumbar disc herniation established by the Spinal Surgery Group and Orthopedic Rehabilitation Group of the Chinese Society of Orthopaedics. Specifically, all participants underwent CT or MRI examinations, and the diagnosis of LDH was confirmed by combining these imaging results with the patient’s medical history, symptoms, and physical signs. This methodology ensures a more accurate and reliable identification of LDH cases.

As mentioned in our Methods section (lines 98-99):

"The diagnosis of LDH was made by combining medical history with the diagnostic criteria proposed by the Spinal Surgery Group and Orthopedic Rehabilitation Group of the Chinese Society of Orthopaedics in 2020. All initial screening subjects were required to undergo CT or MRI examinations, and the diagnosis of LDH was confirmed based on the combination of disease history, symptoms, and physical signs."

We hope this clarification addresses your concerns about the diagnostic approach and reassures you of the validity of our findings.

Thank you once again for your valuable input.

Reviewer #3: 

The authors showed great understanding of the subject matter. The authors showed impressive empirical evidence and makes an original contribution to the subject matter. However, the manuscript need to well written to make it easier to read.

Reply: Thank you for your positive assessment and constructive feedback. We are pleased to hear that you found our understanding of the subject matter and the empirical evidence presented to be strong and original.

We have taken your comments regarding the writing quality very seriously and have revised the manuscript to improve readability and clarity. We believe these changes will make the article more accessible and easier to follow.

Thank you once again for your valuable input and for helping us enhance the quality of our manuscript.

We extend our sincere gratitude to the reviewers for their meticulous examination of our manuscript and for offering insightful feedback that has substantially enhanced its quality. We have thoroughly considered all comments and have implemented the necessary revisions to address the concerns raised. We trust that these modifications have sufficiently addressed the reviewers' suggestions and we welcome any further comments or recommendations.

Thank you again for your time and effort in reviewing our work. We believe that the revised manuscript now provides a more accurate and comprehensive presentation of our research.

Sincerely,

Jihong Hu

---

## [Editor Report · Decision Letter 1]

3 Sep 2024

Prevalence of Lumbar disc herniation and its associated factors: a cross-sectional study in Gansu

PONE-D-24-11202R1

Dear Jihong Hu,

We’re pleased to inform you that your manuscript has been judged scientifically suitable for publication and will be formally accepted for publication once it meets all outstanding technical requirements.

Kind regards,

Md. Feroz Kabir, BPT, MPT, MPH, BPED, MPED

Academic Editor

PLOS ONE
---

## [Editor Report · Acceptance letter]

17 Oct 2024

PONE-D-24-11202R1 

PLOS ONE

Dear Dr. Hu, 

I'm pleased to inform you that your manuscript has been deemed suitable for publication in PLOS ONE. Congratulations! Your manuscript is now being handed over to our production team.

Kind regards, 

on behalf of

Dr. Md. Feroz Kabir 

Academic Editor

PLOS ONE